# Heat Transfer and Pressure Drop of Nanofluid with Rod-like Particles in Turbulent Flows through a Curved Pipe

**DOI:** 10.3390/e24030416

**Published:** 2022-03-16

**Authors:** Wenqian Lin, Ruifang Shi, Jianzhong Lin

**Affiliations:** 1School of Media and Design, Hangzhou Dianzi University, Hangzhou 310018, China; lwq@hdu.edu.cn; 2State Key Laboratory of Fluid Power and Mechatronic System, Zhejiang University, Hangzhou 310027, China; shiruifang@zju.edu.cn; 3Faculty of Mechanical Engineering and & Mechanics, Ningbo University, Ningbo 315201, China

**Keywords:** ZnO/water nanofluid, rod-like nanoparticles, friction factor, heat transfer, energy performance evaluation, turbulent pipe flow, curved pipe, numerical simulation

## Abstract

Pressure drop, heat transfer, and energy performance of ZnO/water nanofluid with rodlike particles flowing through a curved pipe are studied in the range of Reynolds number 5000 ≤ *Re* ≤ 30,000, particle volume concentration 0.1% ≤ *Φ* ≤ 5%, Schmidt number 10^4^ ≤ *Sc* ≤ 3 × 10^5^, particle aspect ratio 2 ≤ *λ* ≤ 14, and Dean number 5 × 10^3^ ≤ *De* ≤ 1.5 × 10^4^. The momentum and energy equations of nanofluid, together with the equation of particle number density for particles, are solved numerically. Some results are validated by comparing with the experimental results. The effect of *Re*, *Φ*, *Sc*, *λ*, and *De* on the friction factor *f* and Nusselt number *Nu* is analyzed. The results showed that the values of *f* are increased with increases in *Φ*, *Sc*, and *De*, and with decreases in *Re* and *λ*. The heat transfer performance is enhanced with increases in *Re*, *Φ*, *λ*, and *De*, and with decreases in *Sc*. The ratio of energy PEC for nanofluid to base fluid is increased with increases in *Re*, *Φ*, *λ*, and *De*, and with decreases in *Sc*. Finally, the formula of ratio of energy PEC for nanofluid to base fluid as a function of *Re*, *Φ*, *Sc*, *λ*, and *De* is derived based on the numerical data.

## 1. Introduction

Mechanical and heat transfer properties of fluids flowing in a pipe are of great interest due to their wide industrial applications in chemical, energy, machinery, and other fields. For improving the performance of heat transfer, nanoparticles are added to the base fluid, i.e., nanofluid, to enhance heat transfer characteristics. However, there are different results on the influence of nanoparticles on the friction factor as well as pressure drop in the pipe. Therefore, it is necessary to simultaneously study mechanical and heat transfer properties of a nanofluid.

In many engineering applications, curved pipes are often used because of the requirement of an actual environment. The properties of nanofluid flowing through a curved pipe are different from those flowing through a straight pipe because of the centrifugal force induced from the pipe curvature. Research on the mechanical and heat transfer properties of nanofluid in the curved pipe have been mainly focused on spherical particles in the past decade. For the case of nanofluid with Al_2_O_3_ particles, the Nusselt number (*Nu*), i.e., heat transfer, was enhanced with increases in the Reynolds number (*Re*) and Prandtl number (*Pr*), and the pressure drop (PD) was increased with increases in particle volume concentration (*Φ*) in the pipe with U-bend [1] and in a U-bend heat exchanger [2]. There was an obvious enhancement of heat transfer with increasing *Re* and *Φ* in a curved pipe with triangular cross-section [3] and in a duct of square cross-section [4]. The frictional entropy generation was lower than the thermal entropy generation [5]. Both friction factor and average value of *Nu* were larger than that for pure water in a finned bend tube [6]. A new equation estimating the pressure loss in turbulent regime was formulated [7]. The local and average value of *Nu* was increased with increases in *Φ*, regardless of *Re* in a curved channel; the function of *Φ* on the increase in heat transfer was more remarkable at larger *Re* [8]. For the nanofluid with Fe_3_O_4_ particles, both *Nu* and consumed power were increased with increasing *Φ* in a heat exchanger [9]. For the nanofluid with CuO particles, the increase in *Nu* value was about 18.6% at 0.06% of *Φ* compared to base fluid with a pumping penalty of 1.09 times in a heat exchanger [10].

Although researchers have paid more attention to the mechanical and heat transfer properties of nanofluids with spherical particles, the effect of particle shape on pressure drop and heat transfer has attracted attention. Studies on this issue have mainly been focused on straight pipes. Heat transfer properties were sensitive to particle shape [11,12]. The disk-like graphite particle alignment weakened the interaction between particles and led to the deterioration of convective heat transfer performance [13]. For diamond shaped particles, the convective heat transfer first increased with increasing *Φ* and then decreased with further increasing *Φ* [14]. The best effect of increasing heat transfer was rod-like particles followed by blade-, plate-, and brick-shaped particles, respectively [15]. The thermal conductivity was enhanced to 12% and 18% at 5.0 v% for the spherical and nearly rectangular ZnO particles, respectively, compared to that of the pure fluid [16]. The nanoparticles with platelet shape showed the highest heat transfer development and heat transfer enhanced with increasing particle volume concentration [17]. The velocity, heat transfer property, and energy functions of the nanofluid exhibited significant variations for nanoparticles of blade, cylinder, platelet, and brick shapes [18]. The maximum performance evaluation criterion corresponded to the brick-shaped nanoparticles, whereas the minimum entropy production corresponded to the case of blade-shaped ones [19]. 

Among all non-spherical particles, rod-like particles are the most common. There have been some studies on the mechanical and heat transfer properties of nanofluids with rod-like particles in the past decade. The shear-induced alignment of the particles had a significant influence on the heat transfer properties, where the particle aspect ratio played an important role [20]. The entropy generation and overall heat transfer coefficient for nanofluids with rod-like particles were higher than those with other shaped particles [21,22]. The heat transfer for rod-like particles was much higher than that for spherical particles [23]. The heat transfer was directly proportional to *Re*, *Φ*, and particle aspect ratio in a laminar pipe flow [24].

As summarized above, and to the extent of our knowledge, research on the pressure drop and heat transfer of nanofluids with rod-like particles in turbulent flows through a 90 degree curved pipe has not been reported, although this situation is very common in engineering applications. A nanofluid with rod-like particles flowing through a curved pipe is a very complicated issue. For example, the rod-like particles would generate anisotropic stresses and affect the turbulent intensity, the centrifugal force of the pipe curve would change the migration and rotation of rod-like particles, thus affecting the spatial and orientation distribution of rod-like particles, and the characteristics of the rod-like particles themselves, such as number density, aspect ratio, and diffusivity, not only determine their own motion properties, but also affect the flow characteristics of fluid. Until now, however, how the above factors affect the mechanical and heat transfer properties of the flow is still unknown, and the relevant mechanism needs to be further clarified. Therefore, the aim of this study is to assess the impact of *Re*, *Φ*, *Sc*, *λ*, and *De* on friction factor, heat transfer, and energy performance.

## 2. Model and Equation

### 2.1. Flow of Nanofluid

A nanofluid with rod-like particles with aspect ratio *λ* flows through a 90 degree curved pipe with inner radius *a* and curved radius *R*, as shown in Figure 1, where the cylindrical coordinate system is used. We define curvature *κ* = *a*/*R* and Dean number *De* = *Re*
$\sqrt{k}$.

For incompressible and fully developed turbulent flow of nanofluid with rod-like particles, the instantaneous velocity, pressure, temperature, rate-of-strain tensor, and particle orientation tensor can be separated into mean part and fluctuation part. Substituting these instantaneous quantities into a continuity equation, modified Navier–Stokes equation with an additional term of rod-like nanoparticles, and energy equations, and then finding the average, we have the corresponding averaged equations [25]:
(1)$$\frac{\partial U_{i}}{\partial xi}=0$$
(2)$$\frac{\partial U_{i}}{\partial t}+U_{j}\frac{\partial U_{i}}{\partial x_{j}}=-\frac{1}{\rho_{nf}}\frac{\partial P}{\partial x_{i}}+\frac{\mu }{\rho_{nf}}\frac{\partial^{2}U_{i}}{\partial^{2}x_{j}}-\frac{\partial \bar{{u^{\prime }}_{i}{u^{\prime }}_{j}}}{\partial x_{j}}+\frac{\mu_{a}}{\rho_{nf}}\frac{\partial }{\partial x_{j}}[a_{ijkl}\varepsilon_{kl}-\frac{1}{3}(I_{ij}a_{kl})\varepsilon_{kl}]$$
(3)$$\frac{\partial T}{\partial t}+U_{j}\frac{\partial T}{\partial x_{j}}=(C_{nf}+C_{T})\frac{\partial^{2}T}{\partial x_{j}^{2}}$$
where *U_i_* and *u_i_*′ are the mean and fluctuation velocity of nanofluid, respectively; *P* is the pressure; *T* is the temperature; *ρ_nf_* is the density of nanofluid; *μ* is the fluid viscosity; $\bar{{u^{\prime }}_{i}{u^{\prime }}_{j}}$ is the Reynolds stress; *a_kl_* and *a_ijkl_* are the mean second- and fourth-order tensors of particle orientation, respectively; *ε_kl_* is the mean rate-of-strain tensor; *C_nf_* is the thermal diffusivity coefficient of nanofluid; *C_T_* = 0.1*k*^2^/*ε* (*k* is the turbulent kinetic energy, *ε* is the turbulent dissipation rate) is the eddy thermal diffusivity coefficient; and *μ_a_* is the generalized viscosity coefficient to account for two-particle interactions [26]:(4)$$\mu_{a}=\frac{4\Phi \lambda^{2}\mu }{3\ln(1/\Phi )}\{1-\frac{\ln[\ln(1/\Phi )]}{\ln(1/\Phi )}+\frac{0.6634}{\ln(1/\Phi )}\}$$
where *Φ* is the particle volume concentration, i.e., percentage of particle volume per unit volume of mixture, and *λ* is the particle aspect ratio.

### 2.2. Density and Thermal Diffusivity of Nanofluid

The density *ρ_nf_* and thermal diffusivity *D**_nf_* of nanofluid in Equations (2) and (3) are [27]:(5)$$\rho_{nf}=(1-\Phi )\rho_{f}+\Phi \rho_{p}$$
(6)$$D_{nf}=\frac{k_{nf}}{{\left(\rho C_{p}\right)}_{nf}}$$
where subscripts *f*, *p*, and *nf* mean fluid, particle, and nanofluid, respectively; *k_nf_* is the thermal conductivity, and (*ρC_p_*)*_nf_* is the heat capacitance [28]:(7)$$k_{nf}=k_{f}\left[\frac{\frac{k_{p}}{k_{f}}+K-K\Phi (1-\frac{k_{p}}{k_{f}})}{\frac{k_{p}}{k_{f}}+K+\Phi (1-\frac{k_{p}}{k_{f}})}\right]$$
(8)$${\left(\rho C_{p}\right)}_{nf}=(1-\Phi ){\left(\rho C_{p}\right)}_{f}+\Phi {\left(\rho C_{p}\right)}_{p}$$
where *K* = 2*Φ*^0.2^*λ* is the shape factor.

### 2.3. Probability Density Function and Tensor of Particle Orientation

In Equation (2), the mean second- and fourth-order tensors of particle orientation are [29]:(9)$$a_{ij}=\oint p_{i}p_{j}\psi \left(p\right)dp,\;a_{ijkl}=\oint p_{i}p_{j}p_{k}p_{l}\psi \left(p\right)dp$$
where *p_i_* and *p* are the unit vector of principal axis and orientation vector of the particle, respectively; and *ψ*(*p*) is the mean probability density function of particle orientation and can be used to determine the likelihood of particle orientation falling within a specific range of values. *ψ*(*p*) is given by:(10)$$\begin{array}{l} \frac{\partial \psi }{\partial t}+U_{j}\frac{\partial \psi }{\partial x_{j}}-D_{rB}\frac{\partial^{2}\psi }{\partial p_{j}^{2}}-\omega_{ji}p_{i}\frac{\partial \psi }{\partial p_{j}}+\lambda \varepsilon_{ji}p_{i}\frac{\partial \psi }{\partial p_{j}}-\lambda \varepsilon_{kl}p_{k}p_{l}p_{j}\frac{\partial \psi }{\partial p_{j}}\\ \;\;\;\;\;\;\;\;\;\;\;\;\;\;\;\;\;\;\;\;\;\;\;-\lambda \varepsilon_{kl}\psi p_{k}p_{l}-D_{rI}\frac{\partial^{2}\psi }{\partial p_{j}^{2}}=\alpha_{\psi x}\frac{\partial^{2}\psi }{\partial x_{j}{}^{2}}“\alpha_{\psi p}\frac{\partial^{2}\psi }{\partial p_{j}{}^{2}} \end{array}$$
where ∂/∂*p_j_* is the gradient operator projected onto the surface of unit sphere; *ω_ij_* = (∂*U_j_*/∂*x_i_* − ∂*U_i_*/∂*x_j_*), *D_rI_* is the rotary diffusion coefficient resulted from particle interaction, *D_rI_* = 0.01$\sqrt{2\varepsilon_{{}_{ij}}\varepsilon_{j}{}_{i}}$ for isotropic *D_rI_* [30]; *α_ψx_* = 1.3(5*k*^2^*ν*/3*ε*)^1/2^ and *α_ψp_* = 0.7(4*ε*/15*ν*)^1/2^ (*ν* is the fluid viscosity) are the dispersion coefficient of linear and angular displacement [31]; and *D_rB_* is the Brownian rotary diffusion coefficient [32,33]:(11)$$D_{rB}=\frac{k_{b}T}{\sqrt{{\left[\frac{3.84\pi \mu L_{p}^{3}(1+0.677/\lambda -0.183/\lambda^{2})}{\lambda^{2}}\right]}^{2}+{\left[\frac{\pi \mu L_{p}^{3}}{3(\ln\lambda -0.662+0.917/\lambda -0.05/\lambda^{2})}\right]}^{2}}}$$
where *k_b_* is the Boltzmann constant; *L_p_* is the particle length.

### 2.4. Turbulent Model

In the present study, the *k*-*ε* turbulent model is used for the range of 5000 ≤ *Re* ≤ 30,000. So, the Reynolds stress $-\rho \bar{{u^{\prime }}_{i}{u^{\prime }}_{j}}$ in Equation (2) is:(12)$$-\rho_{nf}\bar{{u^{\prime }}_{i}{u^{\prime }}_{j}}=2\mu_{T}(\frac{\partial U_{i}}{\partial x_{j}}+\frac{\partial U_{j}}{\partial x_{i}})-\frac{2}{3}\rho_{nf}k\delta_{ij}$$
where *μ_T_* = 0.09*ρ_nf_*
*k*^2^/*ε*, and *k*-equation and *ε*-equation are [34]:(13)$$\rho_{nf}U_{j}\frac{\partial k}{\partial x_{j}}=-\rho_{nf}\bar{{u^{\prime }}_{i}{u^{\prime }}_{j}}\frac{\partial U_{i}}{\partial x_{j}}-\rho_{nf}\varepsilon +\frac{\partial }{\partial x_{j}}[(\mu_{a}+\mu_{T})\frac{\partial k}{\partial x_{j}}]+\rho_{nf}S_{k}$$
(14)$$\rho_{nf}U_{j}\frac{\partial \varepsilon }{\partial x_{j}}=-1.44\frac{\varepsilon }{k}\rho_{nf}\bar{{u^{\prime }}_{i}{u^{\prime }}_{j}}\frac{\partial U_{i}}{\partial x_{j}}-1.92\rho_{nf}\frac{\varepsilon^{2}}{k}+\frac{\partial }{\partial x_{j}}[(\mu_{a}+\frac{\mu_{T}}{1.3})\frac{\partial \varepsilon }{\partial x_{j}}]+\rho_{nf}S_{\varepsilon }$$
where *S_k_* and *S_ε_* are the source terms resulting from the rod-like particles:(15)$$S_{k}=\frac{\mu_{a}}{\rho_{nf}}\left\{\bar{{u^{\prime }}_{n}\frac{\partial }{\partial x_{j}}\left(a_{njkl}{\varepsilon^{\prime }}_{kl}\right)}-\frac{1}{3}I_{nj}\left[\bar{{u^{\prime }}_{n}\frac{\partial }{\partial x_{j}}\left(a_{kl}{\varepsilon^{\prime }}_{kl}\right)}\right]\right\}$$


(16)
$$\begin{array}{l} S_{\varepsilon }=2(\frac{\mu_{a}}{\rho_{nf}}{)}^{2}\left\{\left(\frac{\partial a_{ijkl}}{\partial x_{j}}\bar{\frac{\partial {\varepsilon^{\prime }}_{kl}}{\partial x_{m}}\frac{\partial {u^{\prime }}_{i}}{\partial x_{m}}}\right)+\left(\bar{{\varepsilon^{\prime }}_{kl}\frac{\partial {u^{\prime }}_{i}}{\partial x_{m}}}\frac{\partial^{2}a_{ijkl}}{\partial x_{m}\partial x_{j}}\right)\right.+\left(\frac{\partial a_{ijkl}}{\partial x_{m}}\bar{\frac{\partial {u^{\prime }}_{i}}{\partial x_{m}}\frac{\partial {\varepsilon^{\prime }}_{kl}}{\partial x_{j}}}\right)\\ +\left(a_{ijkl}\bar{\frac{\partial^{2}{\varepsilon^{\prime }}_{kl}}{\partial x_{m}\partial x_{j}}\frac{\partial {u^{\prime }}_{i}}{\partial x_{m}}}\right)-\frac{I_{ij}}{3}\left(\frac{\partial a_{kl}}{\partial x_{j}}\bar{\frac{\partial {\varepsilon^{\prime }}_{kl}}{\partial x_{m}}\frac{\partial {u^{\prime }}_{i}}{\partial x_{m}}}\right)-\frac{I_{ij}}{3}\left(\frac{\partial^{2}a_{kl}}{\partial x_{m}\partial x_{j}}\bar{{\varepsilon^{\prime }}_{kl}\frac{\partial {u^{\prime }}_{i}}{\partial x_{m}}}\right)\\ \left.-\frac{I_{ij}}{3}\left(\frac{\partial a_{kl}}{\partial x_{m}}\bar{\frac{\partial {u^{\prime }}_{i}}{\partial x_{m}}\frac{\partial {\varepsilon^{\prime }}_{kl}}{\partial x_{j}}}\right)-\frac{I_{ij}}{3}\left(a_{kl}\bar{\frac{\partial^{2}{\varepsilon^{\prime }}_{kl}}{\partial x_{m}\partial x_{j}}\frac{\partial {u^{\prime }}_{i}}{\partial x_{m}}}\right)\right\} \end{array}$$


### 2.5. Equation of Particle Number Density

In order to obtain the particle volume concentration *Φ* included in Equations (4)–(8), the particle number density *n* (number of particles contained in volume *v*) should be calculated in advance. Expressing the instantaneous velocity and particle number density as the sum of mean part and fluctuation part, substituting these instantaneous quantities into the equation of number density, and then finding the average, we have [35]:(17)$$\frac{\partial n(v)}{\partial t}+U_{j}\frac{\partial n(v)}{\partial x_{j}}-\frac{\partial }{\partial x_{j}}D_{tB}\frac{\partial n(v)}{\partial x_{j}}-\nu_{t}\frac{\partial^{2}n(v)}{\partial x_{j}{}^{2}}=0$$
where *n*(*v*) is the mean particle number density, *ν_t_* = 0.09*k*^2^/*ε*, and *D_tB_* is the Brownian translational diffusion coefficient [32,33]:(18)$$D_{tB}=\frac{k_{b}T}{\sqrt{{\left[\frac{2\pi \mu L_{p}}{\ln\lambda -0.207+0.980/\lambda -0.133/\lambda^{2}}\right]}^{2}+{\left[\frac{4\pi \mu L_{p}}{\ln\lambda +0.839+0.185/\lambda +0.233/\lambda^{2}}\right]}^{2}}}$$

Solving Equation (17) to obtain *n*(*v*) and then multiplying *n*(*v*) by *v*, we have:(19)$$\int_{0}^{\infty }vn(v)dv=V$$
where *V* is particle volume, and *Φ* can be calculated based on *V*.

### 2.6. Pressure Drop and Nusselt Number

In order to compare with the available results, the friction factor *f* proportional to the pressure drop is given:(20)$$f=\frac{\Delta p}{\rho_{nf}(L/2a)(U_{Sa}^{2}/2)}$$
where ∆*p* is the pressure drop, *L* is the arc length of the curved pipe, *a* is the inner radius of the pipe, and *U_Sa_* is the average velocity of nanofluid in the flow direction.

The Nusselt number is defined as the ratio of heat convection to heat conduction:(21)$$Nu=\frac{{\left.2a\frac{\partial T}{\partial r}\right|}_{r=\pm a}}{\left(T_{w}-T_{m}\right)}$$
where *T_w_* is the wall temperature, and *T_m_* is the mean temperature over cross-section.

## 3. Numerical Method and Parameters

### 3.1. Main Steps

(1)Solving Equations (1)–(4) and (12)–(14) with *Φ* = *μ_a_* = *S_k_* = *S**_ɛ_* = 0 to obtain *U_j_*, *P*, *k*, *ε* and $\bar{{u^{\prime }}_{i}{u^{\prime }}_{j}}$.(2)Solving Equations (17)–(19) to obtain *n*(*v*) and *Φ*.(3)Substituting *Φ* into Equations (4)–(8) to obtain *μ_a_*, *ρ_nf_*,*D_nf_*, *k_nf_* and (*ρC_p_*)*_nf_*.(4)Substituting *U_j_*, *k*, *ε* and Equation (11) into Equation (10) and solving it to obtain *ψ*.(5)Substituting *ψ* into Equation (9) to get *a_ij_* and *a_ijkl_*.(6)Substituting *ρ_nf_*, *μ_a_*, *a_kl_*, *a_ijkl_* and *D_nf_* into Equations (1)–(4) and (12)–(14) to obtain *U_j_*, *P*, *k*, *ε*, $\bar{{u^{\prime }}_{i}{u^{\prime }}_{j}}$ and *T*.(7)Repeating steps (2) to (6) using the new values of *U_j_*, *P*, *k*, *ε*, $\bar{{u^{\prime }}_{i}{u^{\prime }}_{j}}$, and *T* until the difference between the successive results of *U_i_*, *p*, and *T* is less than a definite value.(8)Calculating the friction factor *f* and Nusselt number using Equations (20) and (21).

### 3.2. Numerical Method

A finite volume method is used to solve Equations (1)–(3) and (10)–(18). This method has two major advantages. One is that it enforces the conservation of mass, momentum, and energy at discretized level, and fluxes between adjacent control volumes are directly balanced. The accuracy of conservative schemes is generally higher than that of non-conservative ones. The other is that this method takes full advantage of arbitrary meshes to approximate complex geometries. The SIMPLE [36] and power-law scheme are used to handle the convection term and velocity–pressure coupling term. A staggered mesh and an alternating direction implicit method are employed to solve the discretized equations, i.e., P, T, and U_s_ are located at the center of the meshes, whereas *U*_r_ and *U*_θ_ are located at the mesh lines. The no-slip condition is applied on the walls, and the standard wall function is employed, and the distance between the first mesh center and wall is laid at y^+^ = 30. Equation (9) is integrated by the Simpson formula. The in-house code is used in the numerical simulation. 

### 3.3. Parameters in Numerical Simulation

The nanofluid is a mixture composed of water and ZnO nanoparticles, with a uniform temperature T_0_ = 293 K. For water, ρ_f_ = 998.3 kg/m^3^, C_p_ = 4180 J/kg·K, k_f_ = 0.602 W/m·K, and μ_f_ = 1.005 × 10^−3^ Pa·S. For ZnO particles, ρ_p_ = 5606 kg/m^3^, C_p_ = 520 J/kg·K, and k_p_ = 90 W/m·K. The Boltzmann constant k_b_ is 1.38 × 10^−23^ J/K. We choose ZnO as the nanofluid because it is insoluble in water and has good dispersion and stability in water. The parameter values given above are actual values. The value of dimensionless parameters given in numerical simulation comes from the range of application in practical application. 

Schmidt number is defined as the ratio between the momentum diffusivity and the mass diffusivity:(22)$$Sc=\frac{\mu }{\rho_{f}D_{p}},D_{p}=\frac{k_{b}T}{3\pi \mu d_{p}}$$
where *D_p_* is the particle diffusion coefficient, and *d_p_* is the equivalent diameter of particles.

### 3.4. Mesh Independence Test

The grid system is composed of 64(*r*) × 48(*θ*) × 112(*S*) = 344,064 grid points. The grid is evenly distributed along the *θ* and *S* directions but is densely distributed near the wall in the *r* direction. Grid independence is tested by changing grids, as shown in Table 1, where a convergence criterion is specified, with all the residual errors being less than 10^−4^.

## 4. Results and Discussion

### 4.1. Validation

In order to validate the numerical model and method, we compare the present numerical result of the viscosity of a nanofluid with ZnO rod-like particles based on Equation (4), with experimental results [37] as shown in Figure 2, and pressure drop with experimental results [7], as shown in Figure 3, where the reason for the differences between experiment and simulation is the nanofluid with Al_2_O_3_ spherical particles was used in the experiment. In addition, the pressure drop at low *Re* deviates highly from the experimental data compared to that at high *Re*, which can be attributed to the fact that the *k*-*ε* turbulent model has higher accuracy when used in the flow with high *Re*. 

### 4.2. Friction Factor

Friction factor is proportional to the pressure drop, as shown in Equation (20). A larger friction factor means that more pumping power is needed under the same conditions.

#### 4.2.1. Impact of Reynolds Number

Figure 4 shows the relationship between friction factor *f* and Reynolds number *Re*. In the figure, the Blasius solution [38] with one-seventh power velocity distribution for pure water in a straight pipe (*f* = 0.3164/*Re*^1/4^) is also given as a comparison. We can see that the values of *f* in the nanofluid are larger than those in pure water. The reason is that the rod-like particles are enforced by the fluid to align with flow direction in the nanofluid, which makes the fluid expend extra energy, resulting in an increase in pressure drop. This conclusion is also obtained in the nanofluid with carbon nanotube additives [39]. The values of *f* are decreased with the increase in *Re* for different particle volume concentration *Φ*, indicating that the law that *f* decreases with the increase in *Re* in pure water does not change for the nanofluid. The magnitude of decrease for *f* is large, in the range of *Re* < 20,000, because the turbulent flow has not yet reached a fully developed state. When *Re* > 20,000, the magnitude of decrease for *f* becomes small, and *f* gradually reaches a stable value with increasing *Re*, which shows that the turbulent flow has reached a fully developed state. The values of *f* for Blasius solution are obviously larger than numerical results with *Φ* = 0% in the laminar flow and transition areas (5000 ≤ *Re* ≤ 10,000) because the calculation accuracy is not high when the Blasius solution is applied to the laminar flow and transition areas. However, the values of *f* for Blasius solution and numerical results with *Φ* = 0% are basically consistent because the Blasius solution is suitable for the flow, which reaches a fully developed turbulent state.

#### 4.2.2. Impact of Particle Volume Concentration

The effect of particle volume concentration *Φ* on *f* is also shown in Figure 4, where the values of *f* increase with increases in *Φ*. This can be attributed to the following reasons. (1) More energy is needed to transport the fluid carrying nanoparticles when *Φ* is large. (2) The density and viscosity of the nanofluid are increased with increasing *Φ*, as shown in Equations (4) and (5), which is responsible for the increase in *f*. The effect of *Φ* on *f* is less obvious at high *Re* because the suppression effect of particles on turbulence is more obvious at high *Re*. The friction factor penalty is small at *Φ* = 0.1%, but large at *Φ* = 1~4%, especially for the case at low *Re*.

#### 4.2.3. Impact of Schmidt Number

Friction factor *f* as a function of Reynolds number *Re* for different Schmidt number *Sc* is shown in Figure 5, where *f* increases with increases in *Sc*. Based on the definition of Schmidt number, as shown in Equation (22), *Sc* is directly proportional to the fluid viscosity *μ* and particle diameter *d_p_*, i.e., a large *Sc* corresponds to a large *μ* or *d_p_*. The value of *f* is large in the flow with large *μ*. The particles with large *d_p_* have large inertia so are more likely to be thrown to the outer wall of the curved pipe by centrifugal force, and the accumulation of particles near the outer wall leads to an increase in *f*. In addition, the force of particles acting on the fluid is closely related to the particle size; the particles with large *d_p_* lead to an increase in the turbulence of the flow as well as *f*.

#### 4.2.4. Impact of Particle Aspect Ratio

Figure 6 shows the relationship between *f* and particle aspect ratio *λ*. Increasing *λ* has two opposite effects. On the one hand, increases in *λ* would enhance the generalized viscosity coefficient *μ_a_*, as shown in Equation (4), which leads to an increase in *f*. On the other hand, for the particles with large *λ*, the alignment phenomenon formed by particles under shear is more obvious, which leads to a decrease in viscosity of the nanofluids, in a manner similar to shear thinning, resulting in a decrease in *f*. As shown in Figure 6, the values of *f* increase with decreasing *λ*, which indicates that the effect of *λ* on decreasing *f* is larger than that on increasing *f* in the parameter range discussed in this paper.

#### 4.2.5. Impact of Dean Number

The relationship between Dean number *De* and *f* is shown in Figure 7, where *f* is increased with increases in *De*. As shown in 2.1, *De* is proportional to the pipe curvature *κ*, and the larger *κ* is, the larger the centrifugal force on the fluid and particles. In other words, a large *De* corresponds to a large centrifugal force.Under a large centrifugal force, the particles are more likely to migrate to the outer wall of the curved pipe, and the secondary flow is stronger, both resulting in an increase in *f*.

### 4.3. Heat Transfer

The following factors have a direct impact on heat transfer, which can be reflected by the Nusseltnumber *Nu*. A larger *Nu* corresponds to more active convection.

#### 4.3.1. Impact of Reynolds Number

The Nusselt number *Nu* as a function of *Re* for different particle volume concentration *Φ* is shown in Figure 8, where *Nu* increases with increases in *Re*, which may be attributed to the following reasons. As *Re* increases, the secondary flow intensity increases, and the random motion of nanoparticles caused by turbulent flow becomes more intense, leading to the enhancement of heat transfer. The consistency of particle orientation becomes worse with increases in *Re* [40], thus increasing particle interaction, which is the main energy pathway in the particles, facilitating convective heat transfer. In addition, the increase in *Nu* gradually decreases with increases in *Re*, showing that the effect of heat transfer caused by turbulence is gradually stabilized when the turbulent flow is fully developed.

#### 4.3.2. Impact of Particle Volume Concentration

The values of *Nu* for *Φ* ≠ 0 are larger than that for *Φ* = 0 in Figure 8, which indicates that the rod-like nanoparticles adding to the base fluid can promote convective heat transfer. There are two reasons for this. One is that the rotation of rod-like particles induced by the difference in velocity at the two ends of the particle creates a disturbance to the flow, facilitating convective heat transfer. Another is that two ends of rod-like particles rotating in the flow experience alternately different temperature in the near-wall and bulk regions; thus, heat can be conducted easily from one end to another end of the highly conductive particles, acting as a heat pump to transfer heat into the fluid.

The values of *Nu* are increased with increasing *Φ*, as shown in Figure 8. For nanofluid, both heat transfer coefficient and thermal conductivity are enhanced with increases in *Φ*. However, *Nu* is directly and inversely proportional to the heat transfer coefficient and thermal conductivity, respectively, so the enhancement of heat transfer coefficient plays a more important role in the parameter range discussed in this paper. Nanoparticles interact more frequently at high *Φ*, which enhances the turbulence intensity and reduces the thickness of thermal boundary layer so as to promote heat transfer. In addition, the orientation of rod-like particles indicates the flow direction and form strip structure at low *Φ*, leading to the least effective pathway for convective heat transfer.

In Figure 8, the impact of *Φ* on *Nu* is more obvious at low *Re* than that at high *Re*. The reason is that the convective heat transfer of the nanofluid is directly related to the Brownian diffusion and turbulent diffusion. The Brownian diffusion of particles is dominant at low *Re*, so diffusion intensity is directly related to *Φ*. As the turbulent diffusion, which has little relationship with *Φ*, is dominant at high *Re*, so the impact of *Φ* is weak.

#### 4.3.3. Impact of Schmidt Number

Figure 9 shows the values of *Nu* as a function of *Re* for different Schmidt number *Sc*. We can see that the values of *Nu* are decreased with increases in *Sc*. As shown in Equation (22), *Sc* is directly and inversely proportional to the fluid viscosity and particle diffusion coefficient, respectively. For the case with large *Sc*, on the one hand, the viscous layer near the walls is intensified, resulting in a smaller temperature gradient near the walls, as well as a smaller value of *Nu*, as shown in Equation (21). On the other hand, the diffusion range is small for the particles with small diffusion coefficients, making the heat transfer worse. In Equation (22), *Sc* is proportional to the particle diameter, the particles with large diameter have weaker random motion, and larger particles correspond to the lower number of particles for a fixed *Φ* and hence have weaker effect of surface area; both of these factors make the heat transfer worse.

#### 4.3.4. Impact of Particle Aspect Ratio

The values of *Nu* as a function of *Re* for different particle aspect ratio *λ* are shown in Figure 10, where the values of *Nu* increase with increasing *λ*. For the particles with large *λ*, on the one hand, the nanofluid viscosity is large when *Φ* remains unchanged, as shown in Equation (4), which results in a reduction in *Nu*, as discussed in Section 4.3.3. On the other hand, the particles rotating around the center of mass could produce larger disturbances to the base flow to facilitate convective heat transfer, and the range of heat conduction is expanded when heat is conducted from one end to the other end of the particle. The effects of the above two factors are opposite, and the result of the competition between the two factors makes heat transfer and *Nu* higher for the particles with larger *λ*.

#### 4.3.5. Impact of Dean Number

Figure 11 shows the values of *Nu* as a function of *Re* for different Dean number *De*. The flow characteristics in a curved pipe are related to the pressure drop, viscous force, inertial force, and centrifugal force, and the last one plays an important role. The maximum velocity in the curved pipe is moved from the pipe center to the outer region under the effect of centrifugal force, which produces a stronger convective mixing between the fluid near the center and on the outer pipe wall, thus strengthening the heat transfer. In addition, centrifugal force would induce and strengthen the secondary flow in which particles are involved and expand the migration range, leading to the enhancement of heat transfer. *De* is proportional to the pipe curvature and hence to centrifugal force, so the values of *Nu* are increased with the increase in *De*. Additionally, *De* is proportional to *Re*, and a larger *De* corresponds to a larger *Re* and thinner laminar sublayer near the wall. The thinner the laminar sublayer is, the greater the temperature gradient is, and the larger the value of *Nu* is.

### 4.4. Energy Performance Evaluation Criterion

The values of *Nu* and *f* for the nanofluids are higher than that for the base fluid, as shown in Figure 4 and Figure 8, respectively. It is necessary to balance the increase in the heat transfer and consumed power for using nanofluids to improve heat transfer more effectively. Therefore, the ratio of heat flow rate transferred to the required pumping power, named the energy performance evaluation criterion (PEC), is defined as [37]:(23)$$\mathrm{PEC}=\frac{(T_{out}-T_{\mathrm{in}})\int_{-a}^{a}2\pi rU_{Sa}{\left(\rho C_{p}\right)}_{nf}dr}{\Delta P\int_{-a}^{a}2\pi rU_{Sa}dr}$$

In which *T*_out_ and *T*_in_ are the temperatures at outlet and inlet, respectively.

#### 4.4.1. Impact of *Re* and *Φ*

The ratio of energy PEC for nanofluid to base fluid is defined as PEC*_nf_*/PEC*_f_*, and the relationship between PEC*_nf_*/PEC*_f_* and *Re* for different *Φ* is shown in Figure 12. It can be seen that PEC*_nf_*/PEC*_f_* is less than 1 at low *Re*, indicating that the difference in the friction factor between the nanofluid and base fluid is larger than that in the Nusselt number. The opposite is true for the case that PEC*_nf_*/PEC*_f_* is larger than 1. In Figure 12, PEC*_nf_*/PEC*_f_* is increased with increases in *Φ*, and the increase rate is roughly the same at the given *Φ*. The point of PEC*_nf_*/PEC*_f_* = 1 shifts to low *Re* with increasing *Φ*, i.e., *Re* ≈ 18,000 for *Φ* = 0.1% to *Re* ≈ 12,500 for *Φ* = 5%. It can be inferred that it is better to apply nanofluids to enhance heat transfer at higher *Re* and *Φ* from a comprehensive point of view.

#### 4.4.2. Impact of *Sc*, *λ* and *De*

Figure 13, Figure 14 and Figure 15 show the PEC*_nf_*/PEC*_f_* as a function of *Re* for different *Sc*, *λ*, and *De*, respectively. It can be seen that PEC*_nf_*/PEC*_f_* is increased with increasing *λ* and *De*, and with decreasing *Sc*. The point of PEC*_nf_*/PEC*_f_* = 1 also shifts to low *Re* with decreasing *Sc* and increasing *λ* and *De*. Thus, it is better to apply nanofluids to enhance heat transfer at low *Sc* and high *λ* and *De*.

#### 4.4.3. Correlation Model

As shown in Figure 12, Figure 13, Figure 14 and Figure 15, PEC*_nf_*/PEC*_f_* is directly proportional to *Re*, *Φ*, *λ*, and *De* and inversely proportional to *Sc*. It is necessaryto build a correlation model relating PEC*_nf_*/PEC*_f_* to *Re*, *Φ*, *Sc*, *λ*, and *De* in order to more effectively describe the effect of these parameters on the energy performance evaluation criterion. Firstly, *Re*, *Φ*, *Sc*, *λ*, and *De* are combined into a dimensionless parameter:(24)$$\xi =\frac{Re\Phi \lambda De}{Sc}$$

Then, a formula relating PEC*_t_*/PEC*_f_* to *ξ* is created based on Equation (24) and numerical data in Figure 12, Figure 13, Figure 14 and Figure 15, as:(25)$${\mathrm{PEC}}_{nf}/{\mathrm{PEC}}_{f}=0.99777+0.00154\xi -1.60845\times 10^{-6}\xi^{2}$$

The energy performance evaluation criterion can be calculated conveniently using Equation (25). Figure 16 shows the numerical data in Figure 12, Figure 13, Figure 14 and Figure 15 and Equation (25) and a fitted curve.

## 5. Conclusions

In order to clarify the effect of Reynolds number *Re*, particle volume concentration *Φ*, Schmidt number *Sc*, particle aspect ratio *λ*, and Dean number *De* on friction factor *f* and Nusselt number *Nu* of ZnO/water nanofluid flowing through a curved pipe, the momentum and energy equations of nanofluid together with the equation of particle number density for nanoparticles in the range of 5000 ≤ *Re* ≤ 30,000, 0.1% ≤ *Φ* ≤ 5%, 10^4^ ≤ *Sc* ≤ 3 × 10^5^, 2 ≤ *λ* ≤ 14, 5 × 10^3^ ≤ *De* ≤ 1.5 × 10^4^ are solved numerically. Some results are validated by comparing the present numerical results with the experimental ones. The main conclusions are summarized as follows:

(1)The values of *f* in nanofluid are larger than that in pure water, and are increased with increases in *Φ*, *Sc*, and *Re*, and with decreases in *Re* and *λ*. The magnitude of decrease for *f* is large and small at *Re* < 20,000 and *Re* > 20,000, respectively.(2)Rod-like nanoparticles added to the base fluid can promote convective heat transfer. Heat transfer performance is enhanced with increasesin*Re*, *Φ*, *λ*, and *De*, and with decreases in *Sc*. The effect of *Φ* on the heat transfer is more obvious at low *Re* than that at high *Re*.(3)The ratios of energy PEC for the nanofluid to the base fluid are increased with increases in *Re*, *Φ*, *λ*, and *De*, and with decreases in *Sc*. Finally, the formula of ratio of energy PEC for nanofluid to the base fluid as a function of *Re*, *Φ*, *Sc*, *λ*, and *De* is derived based on the numerical data.

## Figures and Tables

**Figure 1 entropy-24-00416-f001:**
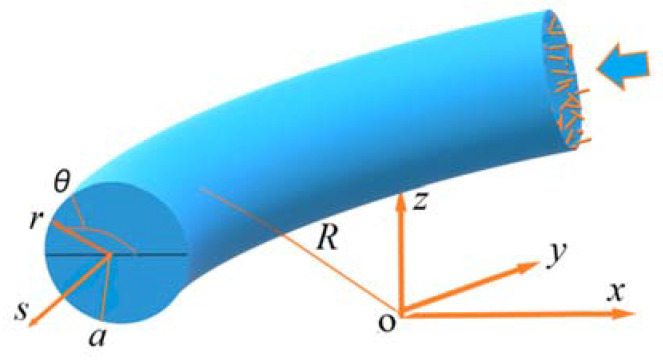
Schematic diagram of nanofluid flowing through a 90 degree curved pipe.

**Figure 2 entropy-24-00416-f002:**
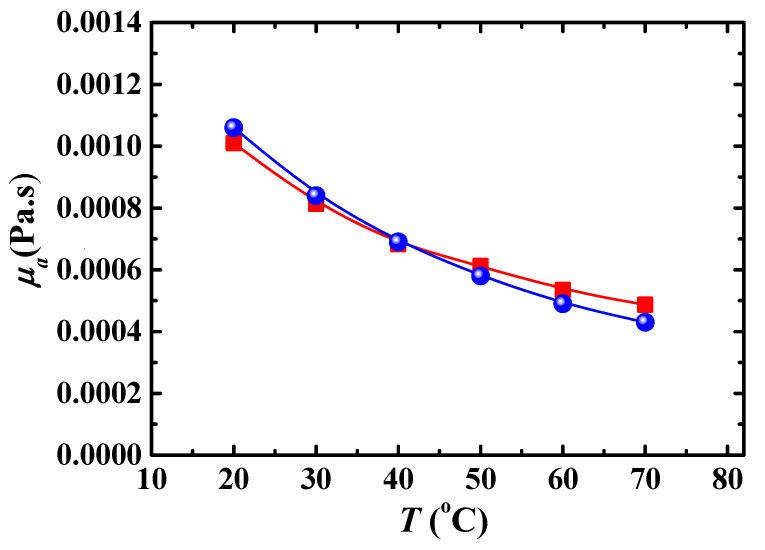
Viscosity of nanofluid with ZnO particles (*λ* = 8, *Φ* = 0.93%). ■: numerical result; ●: experimental result [37].

**Figure 3 entropy-24-00416-f003:**
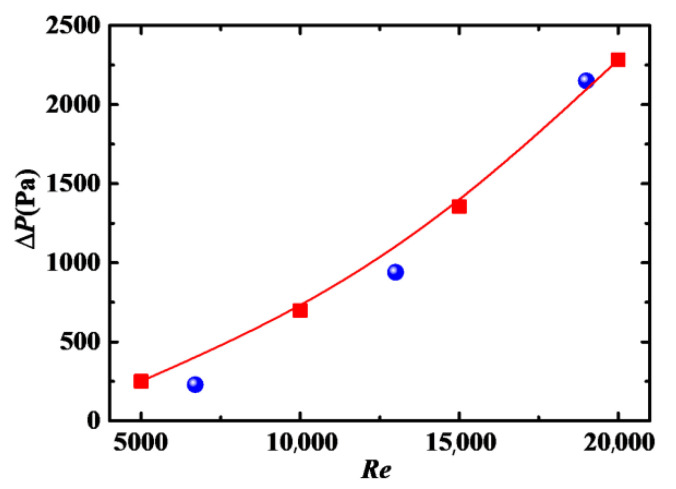
Pressure drop as function of *Re* (*λ* = 1, *Φ* = 2%). ■: present result; ●: experimental result [7].

**Figure 4 entropy-24-00416-f004:**
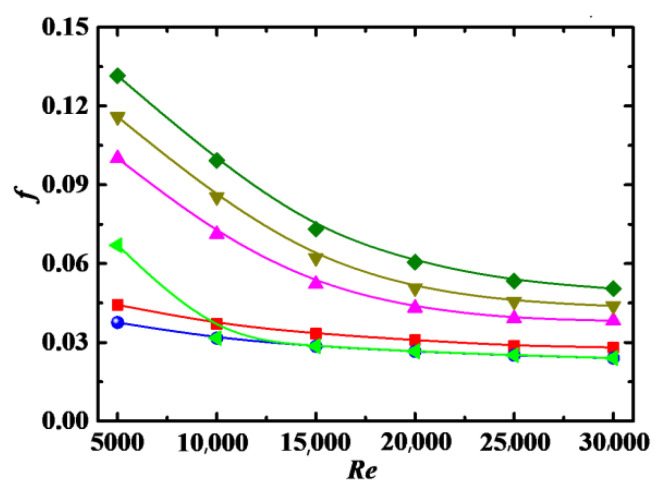
Relationship between *f* and *Re* for different *Φ* (*De* = 1.2 × 10^4^, *Sc* = 10^5^, *λ* = 10). ●: *Φ* = 0%; ◄: Blasius solution (straight pipe) simulation: ■: *Φ* = 0.1%, ▲: *Φ* = 1%, ▼: *Φ* = 3%, ♦: *Φ* = 5%.

**Figure 5 entropy-24-00416-f005:**
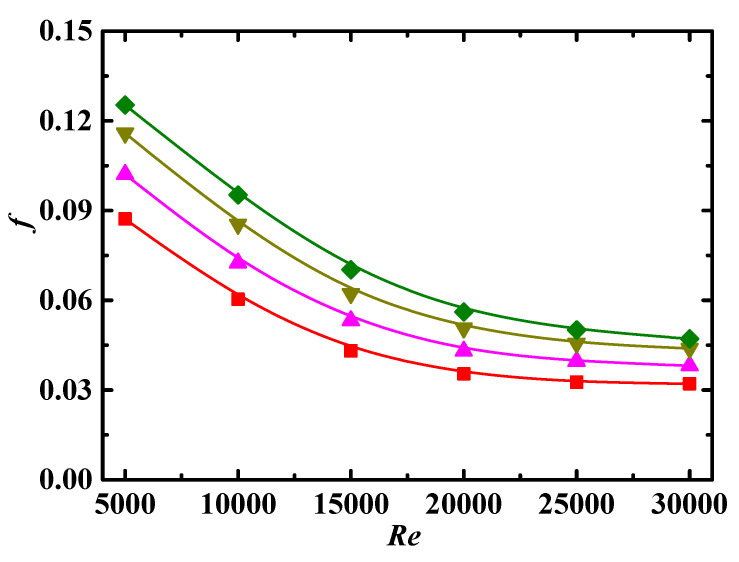
Relationship between *f* and *Re* for different *Sc* (*De* = 1.2 × 10^4^, *Φ* = 3%, *λ* = 10). Simulation: ■: *Sc* = 10^4^, ▲: *Sc* = 5 × 10^4^, ▼: *Sc* = 10^5^, ♦: *Sc* = 3 × 10^5^.

**Figure 6 entropy-24-00416-f006:**
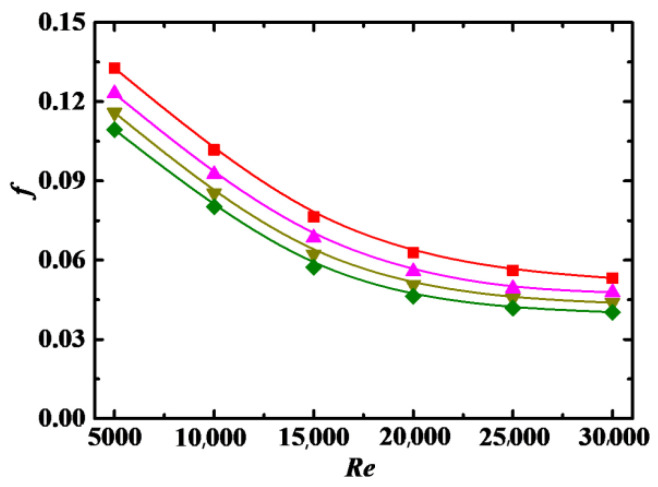
Relationship between *f* and *Re* for different *λ* (*Sc* = 10^5^, *Φ* = 3%, *De* = 1.2 × 10^4^). Simulation: ■: *λ* = 2, ▲: *λ* = 6, ▼: *λ* = 10, ♦: *λ* = 14.

**Figure 7 entropy-24-00416-f007:**
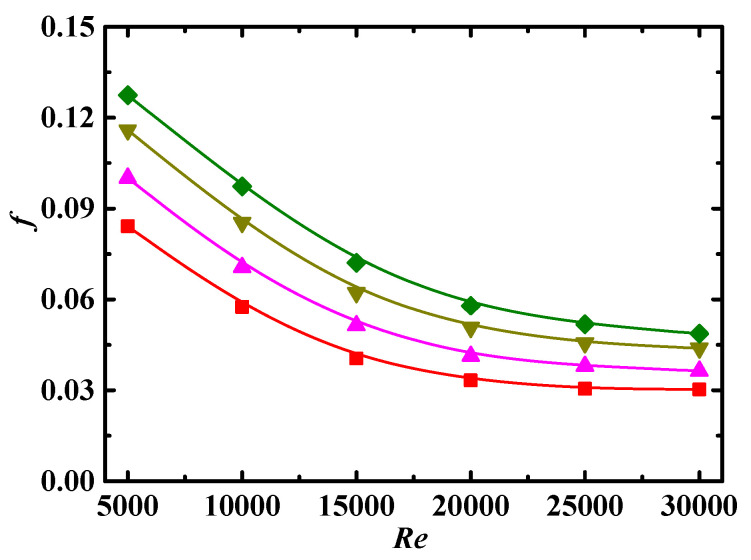
Relationship between *f* and *Re* for different *De* (*Sc* = 10^5^, *Φ* = 3%, *λ* = 10). Simulation: ■: *De* = 5 × 10^3^, ▲: *De* = 9 × 10^3^, ▼: *De* = 1.2 × 10^4^, ♦: *De* = 1.5 × 10^4^.

**Figure 8 entropy-24-00416-f008:**
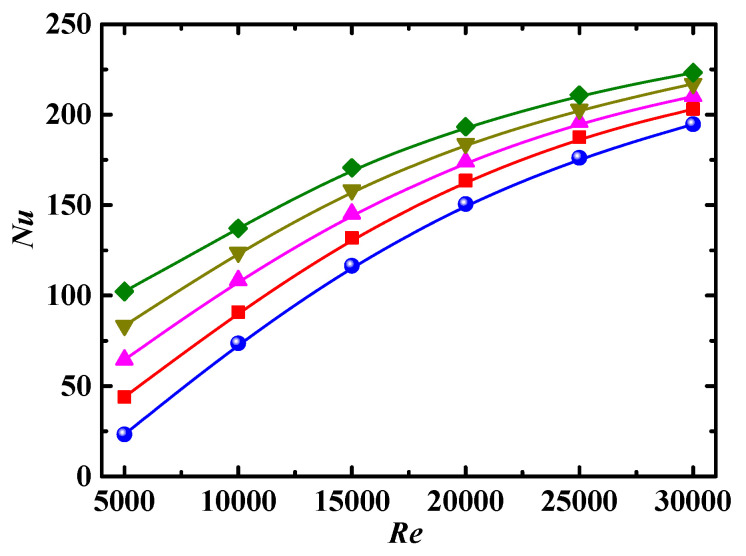
Relationship between *Nu* and *Re* for different *Φ* (*De* = 1.2 × 10^4^, *Sc* = 10^5^, *λ* = 10). Simulation: ●: *Φ* = 0%, ■: *Φ* = 0.1%, ▲: *Φ* = 1%, ▼: *Φ* = 3%, ♦: *Φ* = 5%.

**Figure 9 entropy-24-00416-f009:**
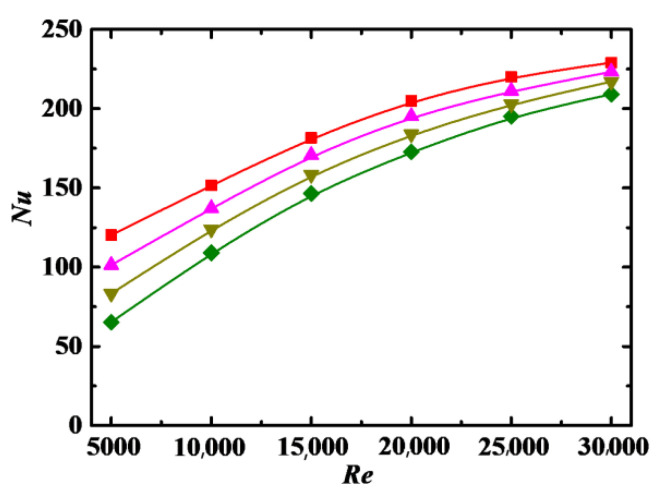
Relationship between *Nu* and *Re* for different *Sc* (*De* = 1.2 × 10^4^, *Φ* = 3%, *λ* = 10). Simulation: ■: *Sc* = 10^4^, ▲: *Sc* = 5 × 10^4^, ▼: *Sc* = 10^5^, ♦: *Sc* = 3 × 10^5^.

**Figure 10 entropy-24-00416-f010:**
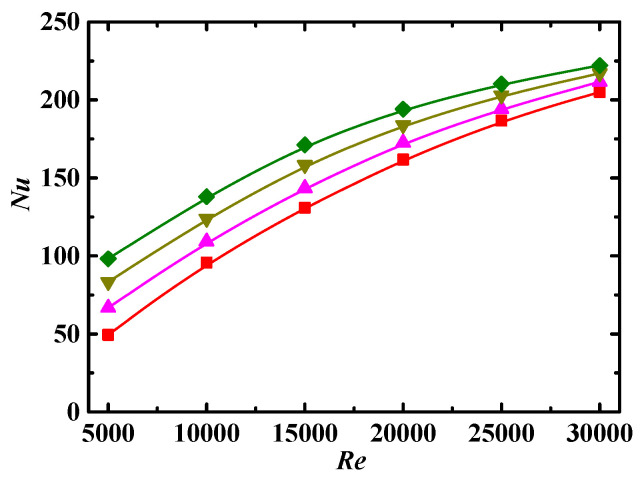
Relationship between *Nu* and *Re* for different *λ* (*Sc* = 10^5^, *Φ* = 3%, *De* = 1.2 × 10^4^). Simulation: ■: *λ* = 2, ▲: *λ* = 6, ▼: *λ* = 10, ♦: *λ* = 14.

**Figure 11 entropy-24-00416-f011:**
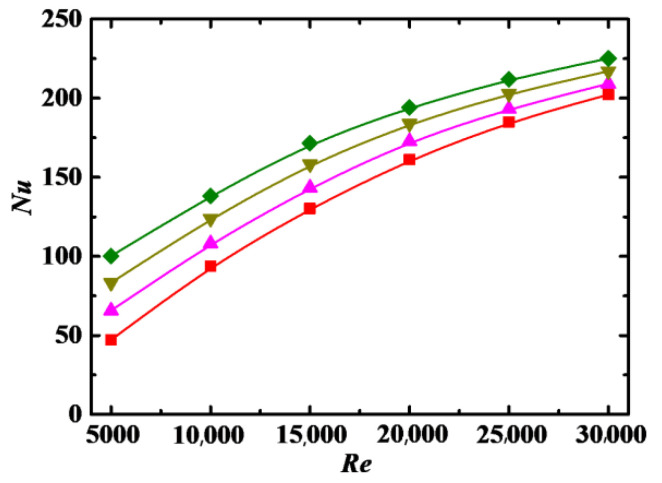
Relationship between *Nu* and *Re* for different *De* (*Sc* = 10^5^, *Φ* = 3%, *λ* = 10). Simulation: ■: *De* = 5 × 10^3^, ▲: *De* = 9 × 10^3^, ▼: *De* = 1.2 × 10^4^, ♦: *De* = 1.5 × 10^4^.

**Figure 12 entropy-24-00416-f012:**
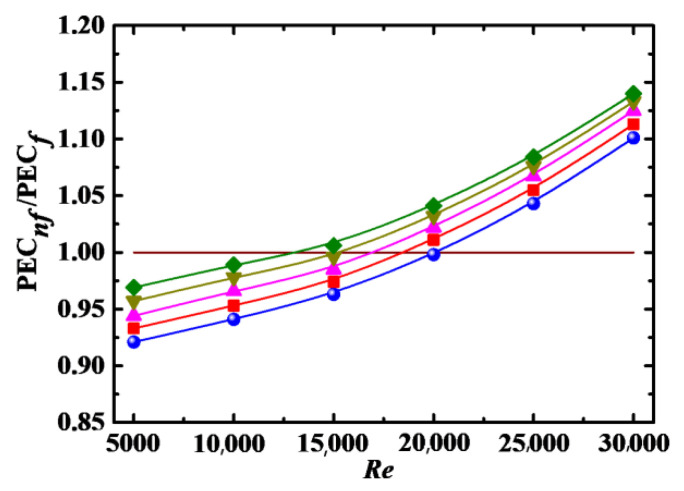
Relationship between PEC*_nf_*/PEC*_f_* and *Re* for different *Φ* (*De* = 1.2 × 10^4^, *Sc* = 10^5^, *λ* = 10). Simulation: ●: *Φ* = 0%, ■: *Φ* = 0.1%, ▲: *Φ* = 1%, ▼: *Φ* = 3%, ♦: *Φ* = 5%.

**Figure 13 entropy-24-00416-f013:**
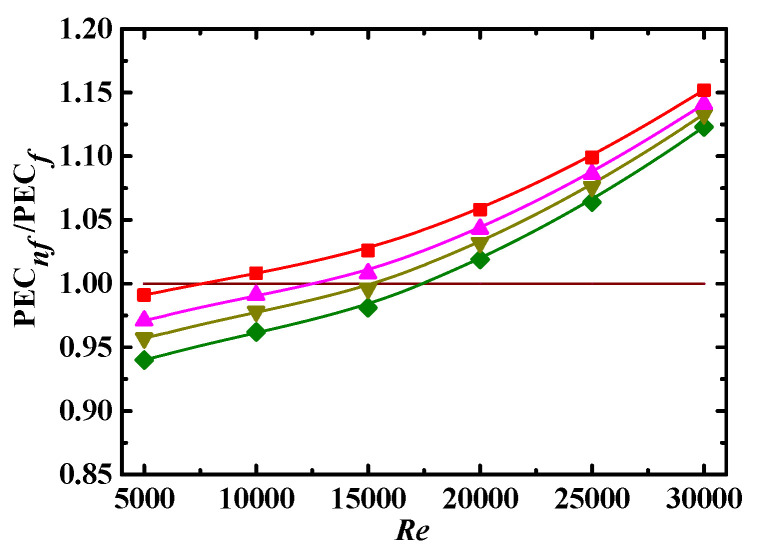
Relationship between PEC*_nf_*/PEC*_f_* and *Re* for different *Sc* (*De* = 1.2 × 10^4^, *Φ* = 3%, *λ* = 10). Simulation: ■: *Sc* = 10^4^, ▲: *Sc* = 5 × 10^4^, ▼: *Sc* =10^5^, ♦: *Sc* = 3 × 10^5^.

**Figure 14 entropy-24-00416-f014:**
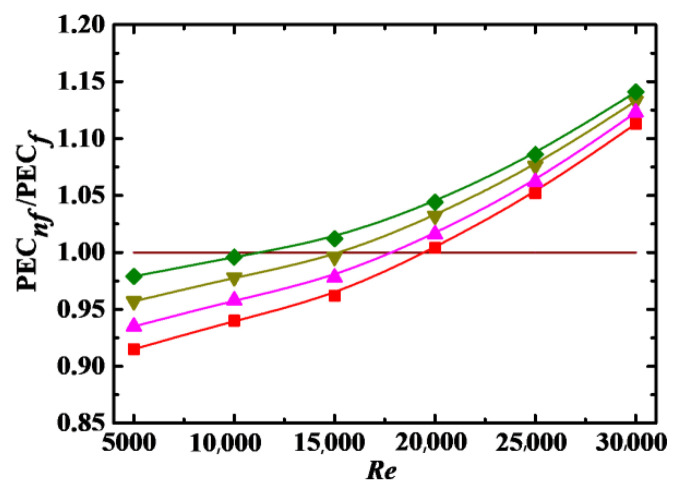
Relationship between PEC*_nf_*/PEC*_f_* and *Re* for different *λ* (*Sc* = 10^5^, *Φ* = 3%, *De* = 1.2 × 10^4^). Simulation: ■: *λ* = 2, ▲: *λ* = 6, ▼: *λ* = 10, ♦: *λ* = 14.

**Figure 15 entropy-24-00416-f015:**
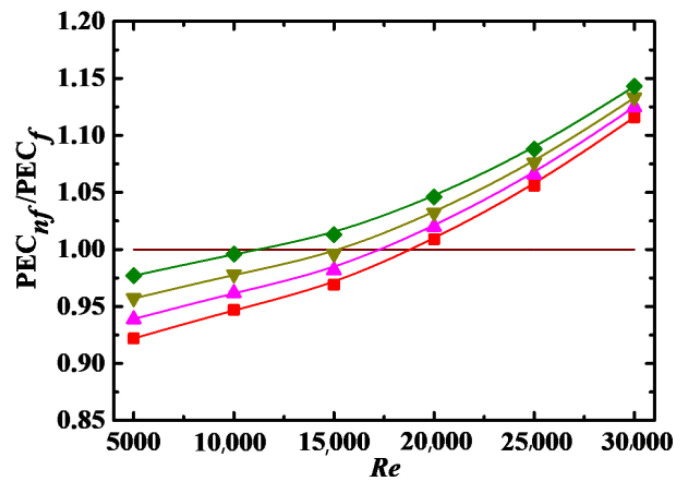
Relationship between PEC*_nf_*/PEC*_f_* and *Re* for different *De* (*Sc* = 10^5^, *Φ* = 3%, *λ* = 10). Simulation: ■: *De* = 5 × 10^3^, ▲: *De* = 9 × 10^3^, ▼: *De* = 1.2 × 10^4^, ♦: *De* = 1.5 × 10^4^.

**Figure 16 entropy-24-00416-f016:**
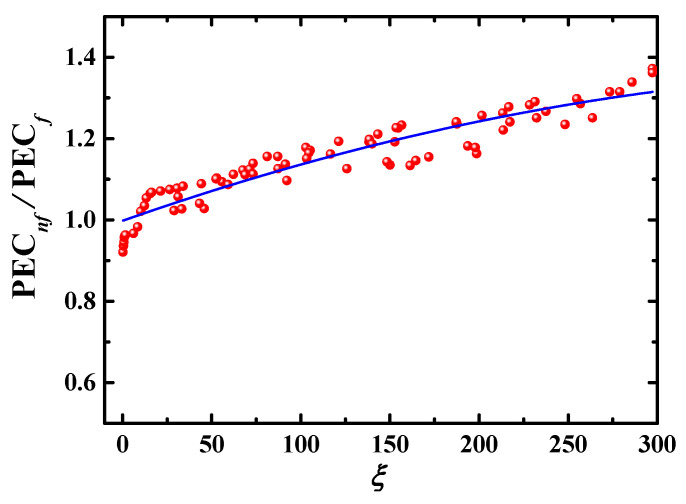
Relationship between PEC*_nf_*/PEC*_f_* and dimensionless parameter *ξ*. ●: numerical data; —: Equation (25).

**Table 1 entropy-24-00416-t001:** Tests on *Nu* when changing grids (*De* = 11,508, *Φ* = 2% *Re* = 20,000).

*r* × *θ* × *S* *Nu*	*r* × *θ* × *S* *Nu*	*r* × *θ* × *S* *Nu*
56 × 48 × 112 177.308	64 × 40 × 112 177.315	64 × 48 × 104 177.301
60 × 48 × 112 177.336	64 × 44 × 112 177.338	64 × 48 × 108 177.333
64 × 48 × 112 177.357	64 × 48 × 112 177.357	64 × 48 × 112 177.357
68 × 48 × 112 177.372	64 × 52 × 112 177.370	64 × 48 × 116 177.374
72 × 48 × 112 177.383	64 × 56 × 112 177.378	64 × 48 × 120 177.386

## Data Availability

Data sharing not applicable.

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
