# Peer review of "Heat Transfer and Pressure Drop of Nanofluid with Rod-like Particles in Turbulent Flows through a Curved Pipe"

_entropy, 2022, doi:10.3390/e24030416_

Round 1

Reviewer 1 Report

Pressure drop, heat transfer and energy performance of Al2O3/water nanofluid with rod-like particles flowing through a curved pipe are studied in the range of Reynolds number 5000≤Re≤30000, particle volume concentration 0.1%≤Φ≤5%, Schmidt number 104≤Sc≤3ï‚´105, particle aspect ratio 2≤λ≤14 and Dean number 5ï‚´103≤De≤1.5ï‚´104. The momentum and energy equations of nanofluid together with the equation of particle number density for particles are solved numerically.

Introduction does not contain a conclusive state of the art with in-depth explanations and examples.

There is confusion between specific heat and thermal diffusivity.

What is the particle number density?

What is the base of this study? What software has been used and how was achieved the model and grid validation? Details about the goodness and similarity of the models are missing. What about the geometry?

Results discussion is very poor, as well as the conclusion section.

Reviewer 2 Report

The work is very interesting and seems to be an original. The paper is well written with a good discussion of the results. The validation of the calculation code gave a good agreement with the experimental results. The work well done but have some  corrections needed to improve the quality of the presentation. I enlist them below

:

  1. Equations 1-3 needs standard references.
  2. What is the importance of Probability density function in your study?
  3. Check the dimensions of Sc and Dp (Equation 22) and  make sure that all are non dimensional
  4. How can be sure the results are valid? More explanation is required
  5. Revise the introduction such that each paragraph shall present the meaning of a concept/keyword. Each paragraph must present at least three theoretical reviews on the same concept/keyword. The introduction section can be supported with some more recent related literature especially since all references are not relatively recent. In this regard I suggest .. Microrotation viscosity effect on turbulent micropolar fluid channel flow, Recurrence quantification analysis of MHD turbulent channel flow
  6. Please discuss at the end of introduction, that why is it important from the aspect of APPLICATION and future experiments to investigate magneto-nanofluid mixed convective flow caused by a vertical permeable plate influenced by nonlinear thermal radiation? This can improve your impact on the field. Please also focus on connecting initial motivation with conclusions at the end. In this format the paper is more of a list of case studies, with parameter analysis and without deeper discussion on how these parameter changes are affecting the topic.
  7. Captions for figures and tables should be checked again. There is a significant lack of information. Please provide readers enough information on them.
  8. Highlight the article innovation point in the abstract. Why should anyone read your article?
  9. Why this numerical method is used for the solution, gives the advantages.
  10. How to choose the values of parameter, explain?

Reviewer 3 Report

In the current work, the authors summarized their work on heat transfer, energy performance, and pressure drop of nanofluids with rod-like particles in turbulent flows through a curved pipe. They were motivated to do so as the work has engineering applications, but no one has conducted it before. In the introduction, the authors illustrated the background and importance of the work.

The authors clearly stated the model core equations, followed by numerical solution steps, and then validated the model results with experimental data where possible. Overall, the manuscript is well written. 

However, I would request the authors a few minor revisions: 

a) Page 56, line 188-189: "The SIMPLE and power-law scheme are used to 

handle the convection term and velocity–pressure coupling term." 

  • please provide a reference. 

b) Page 7, line 224: In Figure 3, model results for the pressure drop at low Reynolds number (Re) deviate highly from experimental data compared to that of at high Re. 

  • Please provide a clear explanation for that phenomenon.

c) Page 7, line 230: Please provide a reference for "Blasius solution".

d) Page 7, line 242-243: "The values of f for straight pipe are obviously larger than that for curved pipe in the laminar flow and transition areas (5000=Re=10000), but very similar for straight and curved pipe in the turbulent flow (Re>10000)."

  • Please provide a reference or explanation for the statement. 

e) Page 11, line 416: In equation (25), the last term (second degree in the polynomial) appears redundant.  

Round 2

Reviewer 1 Report

This is a revised version.

Nevertheless, authors response is not so detailed as expected, as so the changes in the text.

I will come back with several important issues to be considered by the authors.

  1. If it is an in-house code (maybe the self made software is not the appropriate name), please give details about it.
  2. Details about the similarities between current research and models used for validation is mandatory.
  3. What kind of geometry was meshed? The question was not about the dimensionless parameters. What is the envisaged application?
  4. Results discussion in terms of comparison with state of the art is important to validate the results and to explain them.
  5. Introduction is still inconclusive. The added phrase does not solve the quality of the state of the art.
  6. No details about the nanofluid were noticed. It is about rod-like or rot-like? Please correct the mistake and check the entire manuscript! What is actually the nanofluid used in this research and how the properties were evaluated? If the properties were calculated, state clear the goodness and relevance of the results introduced in the code.

Even if some details appear in conclusion, text or abstract (i.e. Al2O3/water nanofluid flowing through a curved pipe or nanofluid with ZnO rod-like particles) it is confusing. It seems that the nanofluid with ZnO was validated with the nanofluid with Al2O3 (section 4.1).

Concluding, there are still many flaws and inconsistencies in this article.

Author Response

see attachment。

Reviewer 4 Report

No comments

Round 3

Reviewer 1 Report

 In my opinion, the manuscript has some major flaws and one of them is validating a nanofluid based on ZnO flow with the help of another nanofluid (alumina). Plus, the properties are not experimental determined.